# Divalent Cation Signaling in *Clostridium perfringens* Spore Germination

**DOI:** 10.3390/microorganisms11030591

**Published:** 2023-02-26

**Authors:** Roua Almatrafi, Saeed Banawas, Mahfuzur R. Sarker

**Affiliations:** 1Department of Biomedical Sciences, College of Veterinary Medicine, Oregon State University, Corvallis, OR 97331, USA; 2Department of Biology, College of Science, University of Bisha, Bisha 61922, Saudi Arabia; 3Department of Medical Laboratories, College of Applied Sciences, Majmaah University, Al-Majmaah 11952, Saudi Arabia; 4Health and Basic Sciences Research Center, Majmaah University, Al-Majmaah 11952, Saudi Arabia; 5Department of Microbiology, College of Science, Oregon State University, Corvallis, OR 97331, USA

**Keywords:** *C. perfringens*, spores, germination, divalent cations, calcium salts

## Abstract

Spore germination plays an essential role in the pathogenesis of *Clostridium perfringens*-associated food poisoning. Germination is initiated when bacterial spores sense various stimuli, including chemicals and enzymes. A previous study showed that dipicolinic acid (DPA) chelated with calcium (Ca-DPA) significantly stimulated spore germination in *C. perfringens*. However, whether Ca^2+^ or DPA alone can induce germination is unknown. Therefore, we aimed to evaluate the possible roles of Ca^2+^ and other divalent cations present in the spore core, such as Mn^2+^ and Mg^2+^, in *C. perfringens* spore germination. Our study demonstrated that (i) Ca-DPA, but not DPA alone, induced *C. perfringens* spore germination, suggesting that Ca^2+^ might play a signaling role; (ii) all tested calcium salts induced spore germination, indicating that Ca^2+^ is critical for germination; (iii) the spore-specific divalent cations Mn^2+^ and Mg^2+^, but not Zn^2+^, induced spore germination, suggesting that spore core-specific divalent cations are involved in *C. perfringens* spore germination; and (iv) endogenous Ca^2+^ and Mg^2+^ are not required for induction of *C. perfringens* spore germination, whereas exogenous and partly endogenous Mn^2+^ are required. Collectively, our results suggest that exogenous spore core-specific divalent cation signals are more important than endogenous signals for the induction of spore germination.

## 1. Introduction

*Clostridium perfringens* is a Gram-positive, encapsulated, obligate anaerobic, spore-forming bacterium that can cause serious diseases in humans and animals [1,2]. Its virulence is due to the production of 20 different toxins [3]. The most common diseases associated with *C. perfringens* type F strains are food poisoning (FP), antibiotic-associated diarrhea in humans, and necrotic enteritis and enterotoxemia in animals [2,4]. *C. perfringens* FP is the second most common bacterial foodborne disease in the USA [1,5]. The symptoms of *C. perfringens* FP—abdominal cramping, vomiting, fever, and diarrhea—begin at 8–18 h after ingestion of contaminated food and typically continue for 12–24 h [2,6]. Although anyone can be infected with *C. perfringens*, the elderly and medically vulnerable individuals experience the most severe infections [2]. While dormant, the spores cannot cause disease; however, once they sense certain environmental signals, the spores germinate and grow [7,8]. Spores that germinate can cause disease; the process begins with spore germination and continues through outgrowth and vegetative cell multiplication [2,6].

To initiate germination, *C. perfringens* spores must contact germinants, such as amino acids, sugars, purine nucleosides, inorganic salts, or a combination thereof [8,9,10]. This is followed by binding to germinant receptors (GRs) located in the inner membrane of the spores, which triggers activation of Csp proteases [9,11,12,13]. These proteases generate an active cortex-lytic enzyme (SleC), which degrades spore cortex peptidoglycan, leading to Ca-DPA release and enabling rehydration of the spore core followed by enzymatic and metabolic resumption [12,13].

Studies have demonstrated a correlation between divalent cations and the success of bacterial spore germination [14,15,16]. In *Bacillus* species, some ions inhibit spore germination, but divalent ions, especially those from DPA-chelated compounds, such as Mg^2+^ and Ca^2+^, increase spore germination [15]. A previous study showed that exogenous Ca-DPA significantly stimulated spore germination in *C. perfringens* [13]. However, it remains unclear whether Ca^2+^ or DPA alone can induce spore germination. In addition, whether DPA or its associated divalent cations can induce germination and the underlying internal and external signaling pathways is yet unknown. In this study, we found that divalent cations at pH 6.0 can stimulate germination without DPA and that exogenous signals are more important than endogenous signals.

## 2. Materials and Methods

### 2.1. Bacterial Strains Used in This Study

Three *C. perfringens* foodborne isolates—SM101, NCTC 10239, and NCTC 8239 [17,18]—were used in this study. They were maintained in cooked meat medium (Difco, BD Diagnostic Systems, Sparks, MD, USA) and stored at −20 °C.

### 2.2. Spore Preparation and Purification

Sporulating cultures of *C. perfringens* were prepared as previously described [9,19]. Briefly, 0.1–0.2 mL of *C. perfringens* cultures in cooked meat medium were inoculated into 10 mL of fresh fluid thioglycolate (FTG) vegetative growth medium and incubated at 37 °C for 24 h [20]. Aliquots (0.4 mL) of the overnight FTG cultures were then inoculated into 10 mL of fresh FTG medium and incubated at 37 °C for 9–12 h [20]. We then inoculated 0.4 mL of these cultures into 10 mL of fresh Duncan-Strong (DS) sporulation medium (1.5% protease peptone, 0.4% yeast extract, 0.1% sodium thioglycolate, 0.5% sodium phosphate dibasic anhydrous, and 0.4% soluble starch); these cultures were incubated at 37 °C for 16–18 h [21]. Spore formation was confirmed using phase-contrast microscopy (Leica MDLS; Leica Microsystems, Wetzlar, Germany). Spores were purified through repeated washing with cold, sterile distilled water and centrifuging (at least 10 times), and then they were sonicated several times for 10 s each until the spore suspensions were 99% free of sporulating cells and debris. Then, we suspended the spores in sterile distilled water, adjusted the optical density at 600 nm (OD_600_) to approximately 6 using a SmartSpec^TM^ 3000 spectrophotometer (Bio-Rad Laboratories, Hercules, CA, USA), and stored the spores at −20 °C until use [9]

### 2.3. Spore Germination

The purified spore suspensions were heat-activated for 10 min at 80 °C and then cooled in a water bath at 21 °C for 5 min. Spore germination was initiated by mixing 33 µL of the heat-activated spore suspension (OD_600_ = 1) with 167 µL of pre-warmed germinant solution (50 mM Ca-DPA (pH 8.0) with CaCl_2_, calcium lactate (C_6_H_10_CaO_6_), Ca(NO_3_)_2_, MgCl_2_, MnCl_2_, MgSO_4_, or ZnCl_2_ (50 mM) adjusted to pH 6.0 with 25 mM Tris HCl buffer; 100 mM KCl (pH 6.0); or 25 mM Tris-HCl buffer (pH 6.0)) in 96-well microtiter plates and incubating the mixtures at 37 °C for 60 min. Spore germination was monitored by measuring the changes in the OD_600_ using a Synergy^TM^ MX multi-mode microplate reader (BioTek^®^ Instruments, Inc., Winooski, VT, USA). A 60% decrease in OD_600_ was defined as complete spore germination, as described previously [9,19]. Spore germination was also confirmed by phase-contrast microscopy (Leica MDLS) at 60 min post-inoculation, as fully germinated spores change from phase-bright to phase-dark. The extent of germination was calculated by measuring the decrease in OD_600_ and is expressed as a percentage of the initial value. Data are from at least two experiments with at least three independent spore preparations.

To examine the effect of pH on the germination rate, germinants were prepared in 25 mM Tris-HCl buffer (pH 6.0) at 37 °C. Similar to previous experiments, all values were obtained from at least two experiments with at least two independent spore preparations. Solutions were prepared at a final concentration of 50 mM in the previously mentioned 25 mM Tris-HCl buffer (pH 6.0). Solutions of the ionic chelators EDTA and EGTA (50 mM) were first prepared in 25 mM Tris-HCl buffer pH 9.0 to speed dissolution and then adjusted to pH 6.0.

### 2.4. Statistical Analysis

Student’s *t*-test was used for comparisons. Statistical significance was set at *p* < 0.05 compared to the initial value (negative control).

## 3. Results

### 3.1. Germination of C. perfringens SM101 Spores in the Presence of Ca-DPA, DPA, and Ca^2+^

Previous studies showed that exogenous Ca-DPA significantly induced spore germination in *Bacillus* and *Clostridium* species [9,19,22,23]. However, whether Ca^2+^ or DPA alone can induce spore germination is unclear. Recent studies on *C. difficile* showed that exogenous Ca^2+^ with the co-germinant taurocholate can efficiently induce spore germination [22]. We assumed that this would also be true for *C. perfringens* spores. To confirm our hypothesis, we first evaluated spore germination of the *C. perfringens* FP wild-type strain SM101 in the presence of 50 mM Ca-DPA (pH 8.0), 50 mM DPA (pH 6.0), or 50 mM CaCl_2_ (pH 6.0). Germination was also carried out with 100 mM KCl (pH 6.0) as a positive control and with 25 mM Tris HCL buffer (pH 6.0) as a negative control.

As expected, *C. perfringens* SM101 spores significantly germinated in the presence of KCl but not in the presence of Tris-HCl. An approximately 60% decrease in OD_600_ was observed when SM101 spores were incubated with 100 mM KCl (pH 6.0) at 37 °C for 60 min (*p* < 0.05) (Figure 1), whereas only a 10%–15% OD_600_ decrease was observed when SM101 spores were incubated with 25 mM Tris-HCl buffer (pH 6.0) under similar conditions (Figure 1). When SM101 spores were incubated with Ca-DPA or CaCl_2_, approximately 40% and 50% decreases in OD_600_ were observed, respectively (*p* < 0.05) (Figure 1). These results indicated that both Ca-DPA and CaCl_2_ were able to induce significant spore germination of *C. perfringens* SM101, although not to the same level as KCl. In contrast, DPA alone did not induce any germination, as no significant difference in OD_600_ was observed between SM101 spores incubated with DPA and SM101 spores incubated with 25 mM Tris-HCl (pH 6.0) (Figure 1). These results were confirmed by phase-contrast microscopy; approximately 80–95% of SM101 spores became phase-dark after 60 min of incubation with KCl, Ca-DPA, or CaCl_2_, whereas approximately 90% of SM101 spores remained phase-bright in the presence of 25 mM Tris-HCL (pH 6.0) or DPA (data not shown). Collectively, these findings indicate that CaCl_2_ alone is sufficient to induce germination of *C. perfringens* SM101 spores.

### 3.2. Germination of C. perfringens FP Isolate Spores with Different Calcium Salts

Initially, we found that, among the compounds tested, 50 mM CaCl_2_ (pH 6.0) induced the highest level of *C. perfringens* SM101 spore germination. To determine whether exogenous Ca^2+^ from other sources can induce SM101 spore germination, spores were incubated with 50 mM CaCl_2_, C_6_H_10_CaO_6_, Ca(NO_3_)_2_, or DPA (pH 6.0); spores incubated with 50 mM Ca-DPA (pH 8.0), 100 mM KCl (pH 6.0), or 25 mM Tris-HCl (pH 6.0) were included as controls.

Our results indicated that SM101 spores germinated, as expected, in the presence of the two positive controls, KCl and Ca-DPA, with 60% and 40% decreases in OD_600_, respectively (*p* < 0.05) (Figure 2A). Under similar experimental conditions, all tested Ca^2+^ ions—i.e., CaCl_2_, Ca(NO_3_)_2_, and C_6_H_10_CaO_6_—induced spore germination, although to different levels, with 50%, 45%, and 35% decreases in OD_600_, respectively (*P* < 0.05) (Figure 2A). As expected, SM101 spores did not germinate in the presence of either buffer (negative control) or DPA alone, with 8% and 10% decreases in OD_600_, respectively (Figure 2A). These results were confirmed by phase-contrast microscopy; approximately 80%–90% of SM101 spores became phase-dark after 60 min of incubation with 50 mM CaCl_2_, Ca(NO_3_)_2_, or C_6_H_10_CaO_6_ at pH 6.0 (data not shown).

To determine whether Ca^2+^ can induce germination of other FP isolate spores, we tested two other FP strains, NCTC 10239 and NCTC 8239, under similar experimental conditions (Figure 2B). Our results showed that spores of both strains germinated when treated with different Ca^2+^ ions, similar to SM101 spores, with 35–40% decreases in OD_600_ compared to the initial OD_600_ values (Figure 2B). These results were confirmed by phase-contrast microscopy; approximately 85–90% of the spore preparations of both FP strains became phase-dark after 60 min of incubation with 50 mM CaCl_2_, Ca(NO_3_)_2_, or C_6_H_10_CaO_6_ (pH 6.0) (data not shown). Collectively, these results suggest that different Ca^2+^ ions can induce germination of most *C. perfringens* FP strains.

### 3.3. Germination of C. perfringens SM101 Spores with Other Spore Core-Specific Divalent Cations

To test whether other spore core-specific divalent cations can induce *C. perfringens* spore germination, we incubated *C. perfringens* SM101 spores with 50 mM MnCl_2_, MgCl_2_, MgSO_4_, or ZnCl_2_. KCl (100 mM, pH 6.0) and 25 mM Tris-HCl (pH 6.0) were included as positive and negative controls, respectively.

Our results showed that *C. perfringens* SM101 spores germinated in the presence of MnCl_2_, MgCl_2_, and MgSO_4_, with 50%, 40%, and 35% decreases in OD_600_ compared to their initial values, respectively (*p* < 0.05) (Figure 3). These results were confirmed by phase-contrast microscopy, which showed that approximately 80–85% of the SM101 spores became phase-dark after 60 min of incubation (data not shown). However, *C. perfringens* SM101 spores did not germinate in the presence of ZnCl_2_ or Tris-HCl buffer, showing only 13% and 10% decreases in OD_600_ compared to their initial values, respectively (Figure 3). These results were confirmed by phase-contrast microscopy; approximately 90% of the SM101 spores remained phase-bright after 60 min of incubation with ZnCl_2_ and Tris-HCl buffer (data not shown). Collectively, these results indicate that spore core-specific divalent cations can trigger spore germination in *C. perfringens*.

### 3.4. Exogenous Ca^2+^ Is Essential for C. perfringens SM101 Spore Germination

As our results showed that exogenous Ca^2+^ alone was able to induce germination of all tested FP isolates (Figure 2A,B), we aimed to understand the role of endogenous Ca^2+^ in germination. To examine whether endogenous or exogenous Ca^2+^ is required to induce spore germination, SM101 spores were incubated with either 100 mM KCl (pH 6.0) or one of the following mixtures: 100 mM KCl + 50 mM EGTA, 100 mM KCl + 50 mM CaCl_2_-EGTA, or 50 mM CaCl_2_-EGTA; EGTA is a calcium-specific chelator (Figure 4).

Our results showed that, in the presence of KCl + EGTA, SM101 spores germinated well, indicating that the release of Ca^2+^ from the spore core was not required to induce spore germination. This was because EGTA chelated the released Ca^2+^ from the spore core and, thus, no free Ca^2+^ was available in the germination solution. In contrast, in the presence of CaCl_2_-EGTA and KCl, we observed a lesser decrease in OD_600_ after 20 min of incubation, indicating that the extra calcium in the presence of EGTA hindered spore germination. Finally, in the presence of CaCl_2_-EGTA, SM101 spore germination was blocked, indicating that extra calcium was needed to initiate spore germination (Figure 4). Collectively, our results indicate that exogenous Ca^2+^, but not endogenous Ca^2+^, is essential for initiating spore germination.

### 3.5. Exogenous Mg^2+^ and Both Exogenous and Endogenous Mn^2+^ Are Essential for C. perfringens SM101 Spore Germination

To examine whether endogenous or exogenous Mg^2+^ is required to induce spore germination, SM101 spores were incubated with 100 mM KCl or one of the following mixtures: 100 mM KCl + 50 mM EDTA, 100 mM KCl + 50 mM MgCl_2_-EDTA, or 50 mM MgCl_2_-EDTA; EDTA is a non-specific metal ion chelator (Figure 5A).

Our results showed that SM101 spores germinated well in the presence of 100 mM KCl + 50 mM EDTA, indicating that the release of Mg^2+^ from the core spores was not required to induce SM101 spore germination. This was because EDTA chelated the released Mg^2+^ from the spore core and, thus, no free Mg^2+^ was available in the germination solution. However, in the presence of 100 mM KCl + 50 mM MgCl_2_-EDTA, SM101 spores stopped germinating after 30 min (shown as a 28% decrease in OD_600_ compared to the initial value). In addition, in the presence of 50 mM MgCl_2_-EDTA, spore germination was completely blocked, indicating that extra magnesium is needed to initiate spore germination. Collectively, our results indicated that exogenous Mg^2^, but not endogenous Mg^2^, is essential for initiating spore germination.

To examine whether endogenous or exogenous Mn^2+^ is needed to induce spore germination, SM101 spores were incubated with 100 mM KCl and one of the following mixtures: 100 mM KCl + 50 mM EDTA, 100 mM KCl + 50 mM MnCl_2_-EDTA, or 50 mM MnCl_2_-EDTA (Figure 5B). Our results indicated that, in the presence of 100 mM KCl + 50 mM EDTA, spores germinated well, as shown by an approximately 40% decrease in OD_600_, meaning that the release of Mn^2+^ from the spore core was not required for the induction of SM101 spore germination because released Mn^2+^ was chelated by EDTA. Surprisingly, in the presence of 100 mM KCl + 50 mM MnCl_2_-EDTA or 50 mM MnCl_2_-EDTA, germination of SM101 spores was almost completely blocked. These results are different from those observed for spore germination with Mg^2+^ and Ca^2+^ and suggest that both exogenous and endogenous Mn^2+^ are essential for the initiation of spore germination.

## 4. Discussion

Previous studies indicated that *C. perfringens* FP isolates can better adapt to environmental changes than other *C. perfringens* isolates [24,25]. In addition, the ability of FP isolates to grow in meat products may be due to the presence of appropriate ions; for example, potassium is found at a higher concentration than sodium, magnesium, manganese, and zinc, although the proportions vary depending on the type of meat [24,26]. Therefore, studying and understanding the germination pathways of FP isolates should help determine the significance and consequences of various ions.

Previous studies have shown that *Bacillus* and *Clostridium* species spores can germinate in ionic media [22,27,28,29]. For *C. difficile*, calcium ions enhance spore germination more than any other divalent or monovalent cation in the presence of taurocholate, which is required as a co-germinant [22]. In addition, potassium ions, sodium ions, and calcium-chelated DPA can trigger the germination of *C. perfringens* spores as either germinants or co-germinants [9,19,30]. Generally, there are a broad range of ionic types that can enhance spore germination, possibly because of the catalytic or physical functions of the ions [27]. However, whether calcium ions or DPA alone can trigger *C. perfringens* spore germination has yet to be clarified. We hypothesized that the divalent cations present in meat products or the spore core might have an important role in spore germination.

The results of this study suggest that spore core-specific divalent cations, but not non-spore specific divalent cations, stimulate spore germination. A possible explanation for this is that divalent cations could be essential for activation of cortex-lytic enzymes, which are important for initiating spore germination [13]. Another explanation could be that some spore core-specific divalent cations, such as Ca^2+^ and Mg^2+^, may be needed to release DPA from the spore core, which is also important for germination [31]. Nevertheless, the effects of various ions on spore germination tend to differ greatly among spore-forming bacteria. For instance, Mn^2+^ enhances the germination of *B. megatherium* [29] and *C. perfringens* spores, whereas it has no effect on *C. difficile* spore germination [22]. We do not yet have a comprehensive explanation for these differences among spore-forming bacteria.

The stages of spore germination in *Bacillus* species are well-studied [7,12]. However, we do not have detailed knowledge about the molecular signals that initiate bacterial spore germination. Therefore, one of the objectives of this study was to identify the endogenous and exogenous signals, specifically the divalent cations. Our results indicate that internal Ca^2+^ and Mg^2+^ are not important signals for germination, but they are required as external signals to induce spore germination. Interestingly, both internal and external Mn^2+^ play important roles in the induction of spore germination. Mn^2+^ may be required for induction of spore germination, given that is required for activation of cortex-lytic enzymes, which are necessary for completing germination. Further studies at both the molecular and protein levels are needed to understand the differences in these ion pathways during germination.

## 5. Conclusions

*C. perfringens* is an anaerobic bacterium that produces metabolically dormant spores that are resistant to environmental stresses and can survive for many years. When environmental conditions are favorable, *C. perfringens* spores germinate and can cause disease. Germination is initiated when bacterial spores sense a chemical signal, which can be a salt, amino acid, cation, or enzyme. DPA chelated with calcium (Ca-DPA) can significantly stimulate germination of *C. perfringens* spores. However, whether Ca^2+^ or DPA alone can induce spore germination has remained unclear. Therefore, in this study, we evaluated the possible roles of Ca^2+^ and other divalent cations present in the spore core (Mn^2+^ and Mg^2+^) in the germination of *C. perfringens* spores.

Our study demonstrated that (i) Ca-DPA, but not DPA alone, induced germination of *C. perfringens* spores, suggesting that Ca^2+^ might have a signaling role in spore germination; (ii) all tested calcium salts, including calcium chloride, calcium carbonate, and calcium nitrate, induced spore germination, indicating that Ca^2+^ ions are critical for this process; (iii) the spore-specific divalent cations Mn^2+^ and Mg^2+^, but not Zn^2+^, could induce spore germination, suggesting that spore core-specific divalent cations are involved in *C. perfringens* spore germination; and (iv) endogenous Ca^2+^ and Mg^2+^ are not required for *C. perfringens* spore germination, whereas exogenous and partly endogenous Mn^2+^ are needed to induce germination.

## Figures and Tables

**Figure 1 microorganisms-11-00591-f001:**
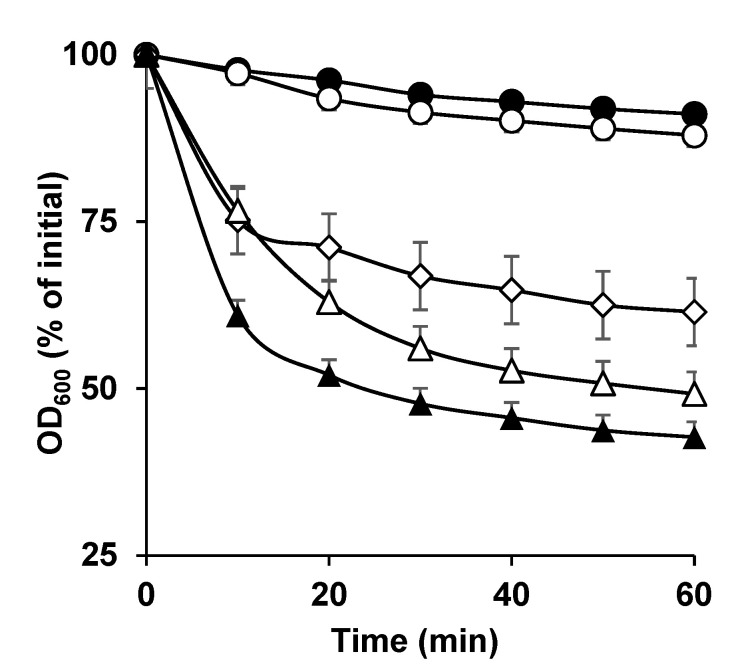
Germination of *C. perfringens* SM101 spores in the presence of DPA, Ca-DPA, and CaCl_2_. Heat-activated spores of strain SM101 were germinated for 60 min at 37 °C in 25 mM Tris-HCl buffer (pH 6.0) (filled circles), 50 mM DPA (pH 6.0) (open circles), 50 mM Ca-DPA (pH 8.0) (open diamonds), 100 mM KCl (pH 6.0) (filled triangles), or 50 mM CaCl_2_ (pH 6.0) (open triangles). Germination was determined by measuring the OD_600_ and calculating the percent decrease compared with the initial values, as described in Section 2.3. Error bars are the standard deviations from the means of at least duplicate experiments with three independent spore preparations.

**Figure 2 microorganisms-11-00591-f002:**
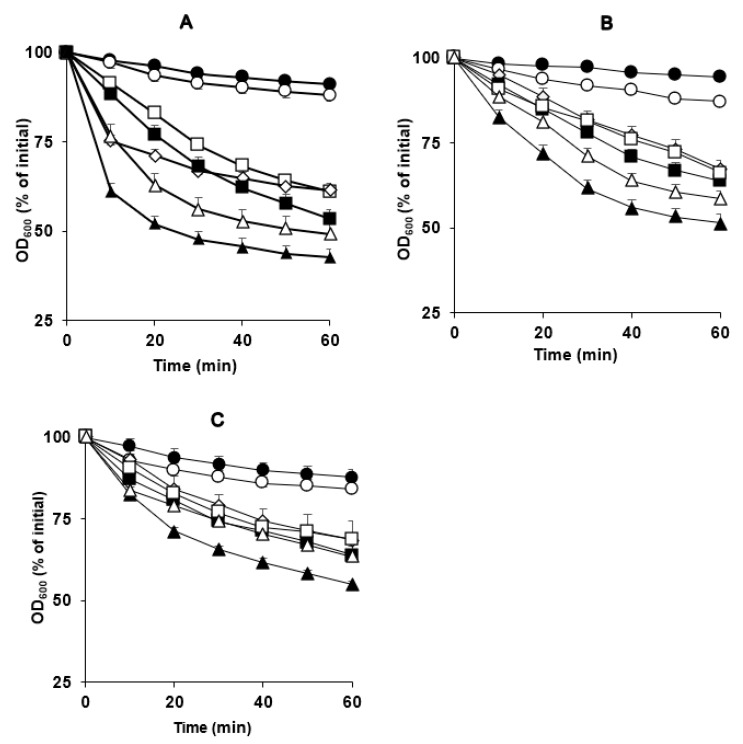
Spore germination of *C. perfringens* strains (**A**) SM101, (**B**) NCTC 10239, and (**C**) NCTC 8239 in the presence of DPA and various Ca^2+^ sources. Heat-activated spores of strains SM101, NCTC 10239, and NCTC 8239 were germinated at 37 °C with 100 mM KCl (pH 6.0) (filled triangles), 50 mM CaCl_2_ (pH 6.0) (open triangles), 50 mM Ca(NO_3_)_2_ (pH 6.0) (filled squares), 50 mM C_6_H_10_CaO_6_ (pH 6.0) (open squares), 50 mM Ca-DPA (pH 8.0) (open diamonds), 50 mM DPA (pH 6.0) (open circles), and 25 mM Tris-HCl (pH 6.0) (filled circles). At each time point, the OD_600_ was measured as described in Section 2.3 and sed to calculate percent germination. Error bars are the standard deviations from the means of at least duplicate experiments with three independent spore preparations. Note that filled squares, open squares, and open diamonds overlap in panel B and open triangles, filled squares, open squares, and open diamonds overlap in panel C.

**Figure 3 microorganisms-11-00591-f003:**
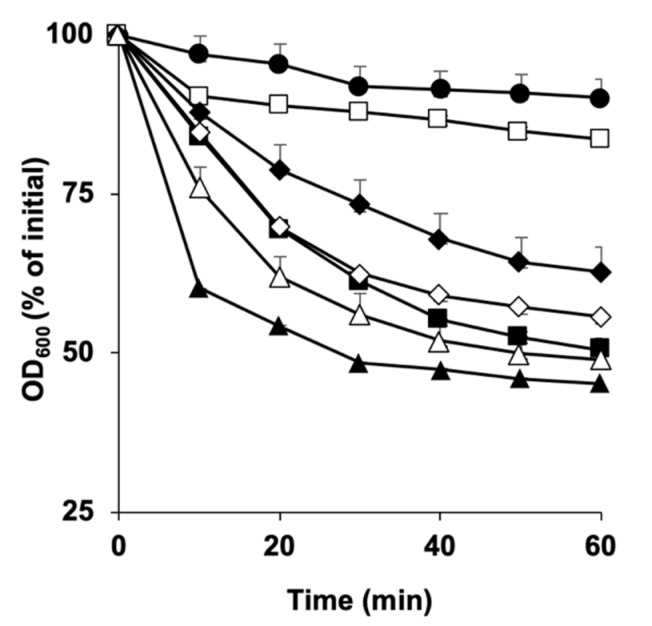
Induction of *C. perfringens* SM101 spore germination by divalent cations. Heat-activated SM101 spores were incubated in 25 mM Tris-HCl buffer (pH 6.0) containing 100 mM KCl (pH 6.0) (filled triangles), 50 mM CaCl_2_ (pH 6.0) (open triangles), 50 mM MnCl_2_ (pH 6.0) (filled squares), 50 mM MgCl_2_ (pH 6.0) (open diamonds), 50 mM MgSO_4_ (pH 6.0) (filled diamonds), or 50 mM ZnCl_2_ (pH 6.0) (open squares) and in 25 mM Tris-HC buffer (pH 6.0) alone (filled circles). The OD_600_ was measured at various time points and used to calculate percent germination. Error bars are the standard deviations from the means of at least duplicate experiments with three independent spore preparations.

**Figure 4 microorganisms-11-00591-f004:**
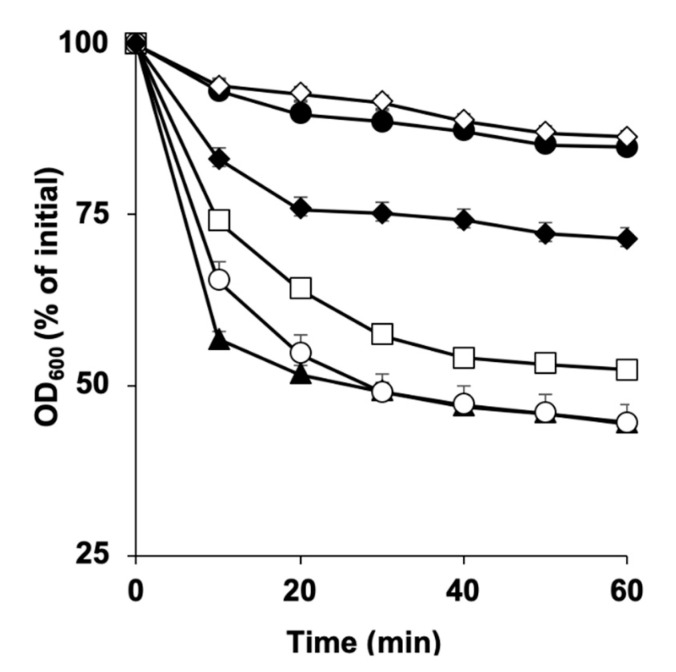
Germination of *C. perfringens* SM101 spores in the presence of Ca^2+^ and a specific Ca^2+^ chelator (EGTA). Heat-activated spores were incubated for 60 min at 37 °C with 50 mM Tris-HCl buffer (pH 6.0) (filled circles), 100 mM KCl (filled triangles), 50 mM EGTA-100 mM KCl (open circles), 50 mM CaCl_2_ (open squares), 50 mM EGTA-CaCl_2_ + 100 mM KCl (filled diamonds), and 50 mM CaCl_2_-EGTA (open diamonds). Then, the OD_600_ was measured at various time points. Error bars are the standard deviations from the means of at least duplicate experiments with two independent spore preparations.

**Figure 5 microorganisms-11-00591-f005:**
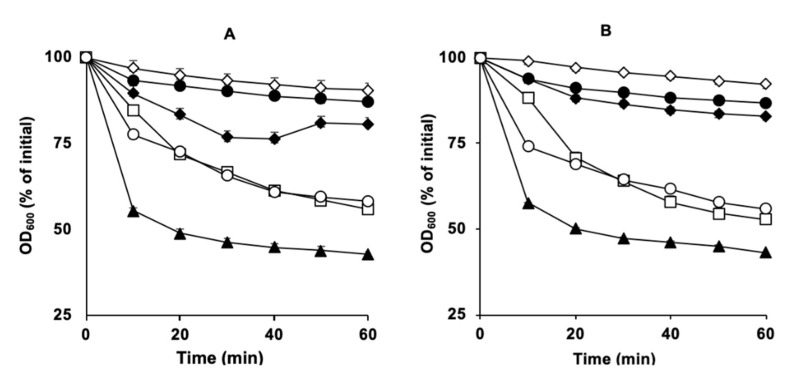
Germination of *C. perfringens* SM101 spores in the presence of MgCl_2_, MnCl_2_, and a general metal ion chelator (EDTA). Heat-activated spores were incubated for 60 min at 37 °C with 50 mM Tris HCl buffer at pH 6.0 supplemented with either (**A**) 100 mM KCl (filled triangles), 50 mM EDTA + 100 mM KCl (open circles), 50 mM MgCl_2_ (open squares), 50 mM EDTA-MgCl_2_ + 100 mM KCl (filled diamonds), 50 mM MgCl_2_-EDTA (open diamonds), or 25 mM Tris HCl buffer (filled circles); or (**B**) 100 mM KCl (filled triangles), 50 mM EDTA + 100 mM KCl (open circles), 50 mM MnCl_2_ (open squares), 50 mM EDTA-MnCl_2_ + 100 mM KCl (filled diamonds), 50 mM MnCl_2_-EDTA (open diamonds), or 25 mM Tris HCl buffer (filled circles). The changes in the OD_600_ were used to calculate the percent germination. Error bars are the standard deviations from the means of at least duplicate experiments with two independent spore preparations.

## Data Availability

The data presented in this study are available on request from the corresponding author.

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
