# Peer review of "Divalent Cation Signaling in Clostridium perfringens Spore Germination"

_microorganisms, 2023, doi:10.3390/microorganisms11030591_

Round 1

Reviewer 1 Report

The ms by Almatrafi et al seeks to separate out the respective roles played in spore germination by divalent cations alone and when they are complexed with DPA. This is an important issue in C. perfringens spore germination and the authors present statistically significant results that indicate the divalent cations are important and the DPA less so.

Major point:

It is not clearly stated, and this reviewer is not sure, how EDTA/EGTA treatment affects the endogenous pool of divalent cations in spores. Since this is used extensively, the authors should clarify this and, if needed, demonstrate with controls how these treatments affect these pool concentrations.

Minor points:

l. 88 and elsewhere: What is C6H10CaO6? This abbreviation for calcium lactate is not commonly known.

l. 150-153: it is stated: “In contrast, DPA-germinated samples heated at 100 °C for 20 min did form colonies on BHI agar at a level similar to that formed by the non-germinated sample, indicating that the spores in the DPA-treated sample remained ungerminated (data not shown).”   It would be helpful to the reader to note how many actual colonies were present, since this is a key statistic for lack of germinating spores.

For Fig. 2B, there are a lot of columns with a similar appearance. It might be easier to read if the X-axis for each column was labeled separately.

Reviewer 2 Report

This paper demonstrated that Ca2+ may be a signaling to C. perfringens spore germination and only certain cations are involved in this event. The exogenous spore core-specific cation signals are more important than endogenous signals for the induction of spore germination. The experiments support these conclusions. Only minor modification needed to make this paper better. 

1) Introduction line 35, type F strains usually cannot cause gas gangrene. 

2) M&M line 101, in all the figure legends the author mentioned three independent repeats for each experiment, but here wrote two repeats. Please make clear how many independent repeats you performed, if only two, please repeat one more time. Only three times independent repeats can support the statistical analysis.

3) Statistical analysis line 117; should provide statistical significance P<0.05 compared to what? the positive or negative control? Please clarified. 

4) Fig 2 please use the identical order in panel (B) bar chart as panel (A), from kCl, cacl2........, which make it easier for the reader to follow.

5)The references need more work: 17) perfringens isolates need small letters; 18) cpe should be italic, also small c; 20) Western should be capital letter; 21) perfringens 1?25)27)28)29) all have some mistakes.
